# A Neural Network Weights Initialization Approach for Diagnosing Real Aircraft Engine Inter-Shaft Bearing Faults



**Tarek Berghout** [1,*] **, Toufik Bentrcia** [1] **, Wei Hong Lim** [2] **and Mohamed Benbouzid** [3,4]

1 Laboratory of Automation and Manufacturing Engineering, University of Batna 2, Batna 05000, Algeria; t.bentrcia@univ-batna2.dz
2 Faculty of Engineering, Technology and Built Environment, UCSI University, Kuala Lumpur 56000, Malaysia; limwh@ucsiuniversity.edu.my
3 Institut de Recherche Dupuy de Lôme (UMR CNRS 6027), University of Brest, 29238 Brest, France; mohamed.benbouzid@univ-brest.fr
4 Logistics Engineering College, Shanghai Maritime University, Shanghai 201306, China
* Correspondence: t.berghout@univ-batna2.dz

**Abstract:** The deep learning diagnosis of aircraft engine-bearing faults enables cost-effective predictive maintenance while playing an important role in increasing the safety, reliability, and efficiency of aircraft operations. Because of highly dynamic and harsh operating conditions of this system, such modeling is challenging due to data complexity and drift, making it difficult to reveal failure patterns. As a result, the objective of this study is dual. To begin, a highly structured data preprocessing strategy ranging from extraction, denoising, outlier removal, scaling, and balancing is provided to solve data complexity that resides specifically in outliers, noise, and data imbalance problems. Gap statistics under k-means clustering are used to evaluate preprocessing results, providing a quantitative estimate of the ideal number of clusters and thereby enhancing data representations. This is the first time, to the best of authors' knowledge, that such a criterion has been employed for an important step in a preliminary ground truth validation in supervised learning. Furthermore, to tackle data drift issues, long-short term memory (LSTM) adaptive learning features are used and subjected to a learning parameter improvement method utilizing recursive weights initialization (RWI) across several rounds. The strength of such methodology can be seen by application to realistic, extremely new, complex, and dynamic data collected from a real test-bench. Cross validation of a single LSTM layer model with only 10 neurons shows its ability to enhance classification performance by 7.7508% over state-of-the-art results, obtaining a classification accuracy of 92.03 ± 0.0849%, which is an exceptional performance in such a benchmark.

**Keywords:** aircraft engine; deep learning; fault diagnosis; inter-shaft bearing; long-short term memory; vibration; weights initialization

## 1. Introduction

Deep learning for the fault diagnosis of aircraft engine bearing is a promising field with immense potential. By using the computational capabilities of deep learning, the way of faults detection and identification for such systems are changed, thus facilitating the exploration of a vast amount of data with remarkable accuracy and speed. Deep learning can ultimately improve aircraft safety, reliability, and efficiency by enabling proactive maintenance and preventing catastrophic failures [1]. Deep learning for such a particular problem, however, faces challenges related to both data complexity and drift. Data complexity occurs when systems generate large volumes of measurements while immersed in a higher level of distortions including an imbalanced number of patterns driven with higher level of noise and outliers, making it difficult for the deep learning models to capture underlying data and their relationships. The impact of data complexity on deep learning models can be mitigated by implementing adequate data preprocessing to address those challenges [2].

On the other hand, data drift refers to continuous change in statistical properties over time. In fault diagnosis, this can occur due to various factors such as variations in operating conditions, equipment degradation, or changes in the underlying dynamics of the system. Accordingly, poor generalization performance may result if the deep learning model is not adaptive. To this end, it is highly recommended to discuss recent advances in this topic, supporting the necessity of the contributions proposed in this paper. Accordingly, this section is dedicated to analyzing related works, uncovering research gaps, and presenting the contributions of this paper while dictating the outlines of the entire paper.

*1.1. Related Work Analysis and Research Gaps*

Related research analysis in this section is carried out based on a significant criterion in order to disclose legitimacy of the paper's proposal (i.e., data preprocessing and RWI adaptive deep learning), uncover research gaps, and demonstrate the necessity of proposed contributions. This means that any reviewed work is expected to address data complexity (e.g., feature extraction, outlier removal, denoising, balancing, and so on) as well as data drift issues (e.g., adaptive learning). Furthermore, because this paper is based on a realistic dataset generated from a real engine test-bench, discussing such an important aspect as an additional analytical criterion would be of paramount importance in uncovering extra gaps in research.

As a result, a set of recent works carried out on aeronautical bearing fault diagnosis topic are selected. The selection procedures are limited to most recent papers published in 2023 in world-renowned databases. Search keywords include deep learning, fault diagnosis, bearing, and aircraft engine. For example, the authors of [3] targeted the presence of noise, class imbalance (i.e., data complexity), and data unavailability issues. This is achieved by considering the use of a domain adaptation approach integrated, respectively, with a meta-bi-classifier gradient divergence method. Their work focuses on improving learning performance by focusing on both approximation and generalization processes and not on data representation specifically as for deep learning. The mathematical formulation of their proposal pays more attention to the learning process and gradient discrepancy while data drift and adaptive learning do not receive much attention. In this case, feature extraction is done automatically using a set of residual classifiers, and problems with class imbalance are resolved through a particular weighting procedure. The authors have employed numerous bearing datasets in order to validate their model. One of these datasets is an aeronautical bearing dataset, which was produced using a test bench system [4]. In [5], the authors investigated a Grampian noise reduction convolutional neural network to deal with the problem of noise presence and data complexity issues. Data complexity is targeted via automatic feature extraction of convolutional filters. While involving the same dataset from [4], the methodology does not discuss outlier detection and removal, data imbalance, data drift or any other data preprocessing paths for reducing data complexity and improving its quality. In [6], authors proposed a correlated feature distribution matching approach to serve as a cross-domain fault diagnosis model. Adaptive learning rules are involved to make sure that the model is up-to-date to any possible data change. Likewise, the same dataset used by previous works is exploited in this case [4]. However, obtained results could suffer from several drawbacks related to model generalizability which resulted due to not taking into account resiliency against data drift problem. In [7], an approach combining successive variational mode decomposition and blind source separation based on slap swarm optimization for bearing fault diagnosis is proposed. The study basically targets data complexity from the power spectrum analysis perspective of improving the noise-to-ratio of vibration signals. A similar dataset that is generally generated from a test-bench experiment is involved in evaluating this methodology. Accordingly, this work is well-thought-out less generalized considering analysis criteria proposed in this section. In [8], an interesting work was carried out on a realistic test-bench aircraft engine. Adaptive learning features of LSTM network are involved in the case. As a result, obtained conclusions are more realistic than those of previously discussed works. Nevertheless, owing to the fact that the

paper aims to discuss data generation process and experiment circumstances as a primary goal, no additional tests are carried out in the context of aforementioned data complexity criteria considering noise and outlier more specifically. This means that accordingly, Table 1 in this case is dedicated to a better illustration of important insights about research gaps in state-of-the-art-literature while using previously discussed criterion to provide further critical conclusion.

**Table 1.** A summary of related works according to the proposed criterion.

| Refs. | Data Complexity | | | | Data Drift | Realistic Test Bed? |
|---|---|---|---|---|---|---|
| | Feature Extraction | Denoising | Outliers Removing | Class Balancing | | |
| [3] | ✓ | ✓ | ✗ | ✓ | ✗ | ✗ |
| [5] | ✗ | ✓ | ✗ | ✗ | ✗ | ✗ |
| [6] | ✗ | ✗ | ✗ | ✓ | ✓ | ✗ |
| [7] | ✓ | ✗ | ✗ | ✗ | ✗ | ✗ |
| [8] | ✗ | ✗ | ✗ | ✗ | ✓ | ✓ |
| This paper | ✓ | ✓ | ✓ | ✓ | ✓ | ✓ |

According to the previous analysis, the following research gaps can be listed.

1. None of the discussed works consider realistic datasets except for reference [8]. Instead, a test-bench is always used to generate these data. In terms of conclusions, the results cannot be projected in real-world circumstances at this moment.
2. Only the works introduced in [6,8] consider the data drift problem. A significant gap in data change research is created by biasing prediction algorithms toward new, unseen data samples.
3. Outlier removals received no attention in this case. In fact, this ignores real operating conditions that always result in massive data distortions, leading to increasing prediction uncertainties.
4. Other data complexity reduction issues related to feature extraction, noise removal, and class imbalance, receive less reasonable attention, but there is undoubtedly a need to consider such constraints since driven sequential data are always exposed to such uncertainty under real-world operational conditions.

*1.2. Contributions*

The contributions of this paper are built on the previous analysis results addressing specifically extracted research gaps. Accordingly, the contributions of this paper can be highlighted as follows:

1. In an attempt to reach more realistic conclusions and generalize obtained results via investigating new unseen samples, a realistic dataset of inter-shaft bearing faults is used in this case [8]. This enables obtained further reliable conclusions compared to ones obtained from non-realistic test-benches experiments in the state of the art by exploring a further challenging feature space emulating real condition.
2. Real systems are usually subject to change either in physical properties (i.e., degradation) or in operating conditions. In this context, our work considers using adaptive learning features of LSTM strengthened by root-mean-square propagation to improve its adaptability and allow better generalization on upcoming data.
3. With the aim of improving data quality, a set of data preprocessing layers are well constructed for this purpose. These layers integrate algorithms for future extraction, outlier removal, denoising, scaling and class balancing with different types to analyze and explore different data features and further improving its scatters representational quality.
4. The results of data preprocessing layers are made subject to final data quality assessment layer. Such investigation is rendered available by involving Gap analysis

under k-means clustering to identify the optimal number of clusters required to group similar data points effectively. Gap analysis helps to assess clustering quality results by comparing the within-cluster dispersion to that of data. The analysis provides insights into determining the appropriate number of clusters, which aids to see whether prepared data patterns could be distinguished or not by the supervised learning algorithm since labels are already existing.

5. A further process of learning parameters initialization is given to the LSTM network in a sort of collaborative learning from a series of best LSTM approximators recursively in multiple rounds via RWI. This is expected to help in reaching better understanding of data drift and allowing the LSTM network to capture better performances rather than random parameter initialization.

### 1.3. Outlines

To guarantee that the introduced claims are clearly illustrated and proved, the roadmap for this article comes in a well-organized structure with enough information to duplicate its underlying findings, as follows. This paper consists of four different sections. In addition to the introductory part of Section 1, Section 2 provides important information about the dataset used in this work, test bench system, preprocessing methodologies and results. Moreover, Section 3 is devoted to proposed methodology description, its application, its results and its discussion. Finally, Section 4 concludes the work with the most pertinent findings besides some future opportunities. It should be mentioned that concerning different mathematical illustrations through the entire text, this paper focuses and limits these illustrations only to introducing novel formulas proposed as main contributions. For other well-known theoretical basics, we introduce a set of standard and original resources to which interested reader can refer. This ensures providing a clear and effective content centered on critical aspects of the work.

## 2. Materials

To better understand the data generation mechanism as well as preprocessing methodology proposed in this work, this section provides both description of the test-bench, data collection and preprocessing in two distinct subsections. It is also worth mentioning that the content of these subsections is limited to necessary details for understanding and reproducing the experiment carried out in this work without delving into any additional theoretical descriptions.

### 2.1. Data Description

A real test-bench system appearing in Figure 1 is used to generate data investigated in this work. It contains a modified dual-spool aircraft engine with removed rotor blades, combustion chamber, and certain accessory casings, installed in costume designed wagon for displacement reasons. The modified aircraft engine is a dual-rotor system driven by two drive motors for both lower pressure (LP) and higher pressure (HP) spools. The LP motor rotates directly the LP spool while a gearbox is used to control speed of HP spool rotation. The structure, including the LP, the HP compressors, in addition to LP and HP turbines are retained. The main load-bearing casing, inter-shaft bearing and five support bearings are also retained in the system. The test bed is supported by a lubricating system ensuring smooth and effective running as in real conditions. The studied type of bearing in this case is an inter-shaft bearing with 15 rolling elements, 30 mm in inner ring diameter, 65 mm in outer ring diameter, 55 mm in pitch circle diameter, 7.5 mm in rolling element diameter, and a nominal pressure angle of $0°$ N.

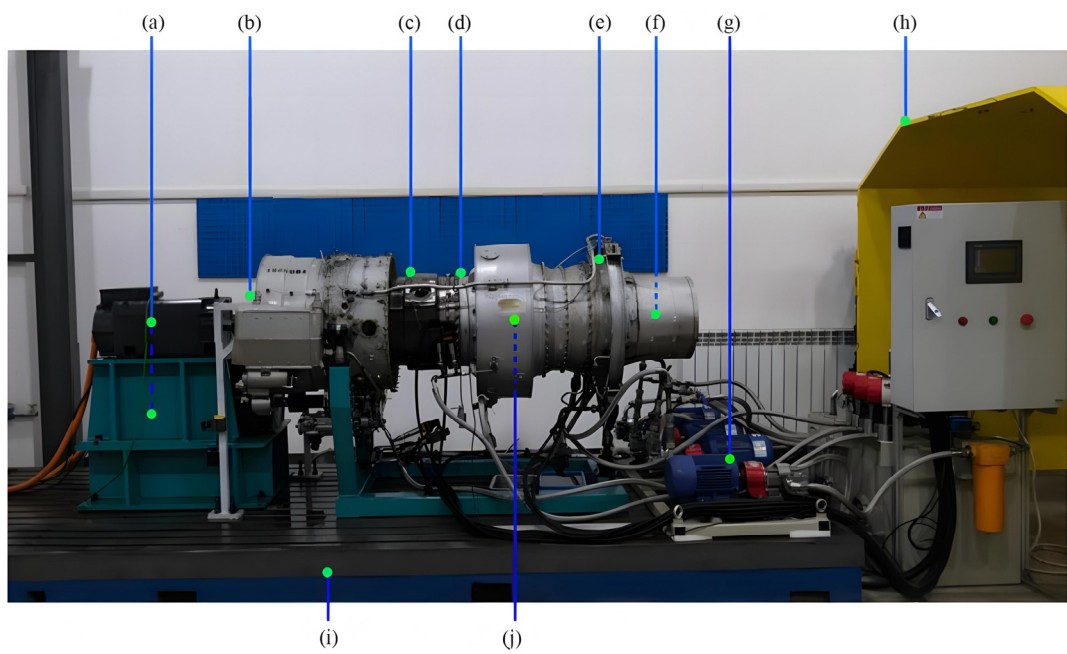

**Figure 1.** A real aircraft engine test bench system: (**a**) the HP and LP motor drive system; (**b**) position of two displacement sensors; (**c–f**) acceleration sensors (i.e., vibration); (**g**) lubricating system; (**h**) laboratory protective shield holding surveillance cameras; (**i**) custom-designed wagon; and (**j**) approximate position of the inter-shaft bearings. Reproduced from: [8], ISTP: 2023, by making changes related to denoising, improving quality, and labeling.

When collecting the dataset used in this work, a clearly defined procedure is followed with five different tests, each of which is linked to specific experimental conditions for both healthy and faulty bearings. These tests, which include parameters such as rotation speed and speed ratio of LP and HP rotors, are referred to as "operating conditions" in this work. Likewise, it is thought about "operating modes" as the normal and fault operating behaviors of the bearings. Bearings fault modes are associated with artificial manipulations of the bearing structure by involving cut length, depth and position as the main parameters. Using installed displacement and acceleration sensors, these five operating condition tests are used to determine 3 different operating mode categories—1 healthy and 2 faulty—(see Figure 1). For unhealthy operating modes, artificial faults are created by a wire cutting causing both inter ring and outer ring faults with different diameters and depths. The process involves sophisticated, well-thought-out and planned assembly and disassembly tasks, as clearly shown in Figures 8–12 from [8]. Accordingly, five sets (i.e., data1–data5) are collected; in particular, 2 sets of healthy operating modes (i.e., data1 and data2), three sets of faults modes (data3–data5) of inter-ring (depth = 0.5 mm, length = 0.5 mm), inner ring (depth = 0.5 mm, length = 0.1 mm), and outer-ring (depth = 0.5 mm, length = 0.5 mm), respectively. All subsets are collected with a sampling rate of 25,000 Hz and stored in a form of 3D time series variable with dimensions of $450 \times 6 \times 20{,}480$ for data2 and data5, and of $504 \times 6 \times 20{,}480$ for data1, data3 and data4.

Concerning LP and HP rotor speeds, each dataset (data1–data5) collected passing by 28 different speed groups, as reflected by curves of Figure 2, while the time separately taken by each group was unrevealed. According to the information collected in the introductory paper of the dataset, there are no specific details to explain these information, which originally prescribed the rotor speed test plan (see Table 3 from [8]). To gain better insight, the curves in Figure 2 were created using values from Table 3 from [8]. Figure 2 appears to offer three different test scenarios. The first scenario belongs to group (0–6), where the speed acceleration of the rotors was increased rapidly to reach about 3500 rpm for HP rotor and 4200 rpm for LP rotor. In this case, the second scenario started from group (7–21)

slightly reduces the speed acceleration compared to the previous one, while the maximum obtained speed was 3000 rpm for the LP rotor and 5000 rpm for the HP rotor. In both scenarios, the speed ratio, which refers to the ratio between LP rotor speed and the HP rotor speed, remained constant. Afterwards, another similar scenario of group (22–28) was carried out with different speed starting points for LP and HP rotors (300 rpm and 3600 rpm, respectively), but with a monotonically increasing speed ratio and constant LP rotor speed. It is worth noting that no specific details about this particular change in speed test plan were provided in the introductory paper. However, as far as we know, the purpose of this plan was to ensure that the test was carried out under controlled conditions and that the data collected were consistent and reliable for later analysis.

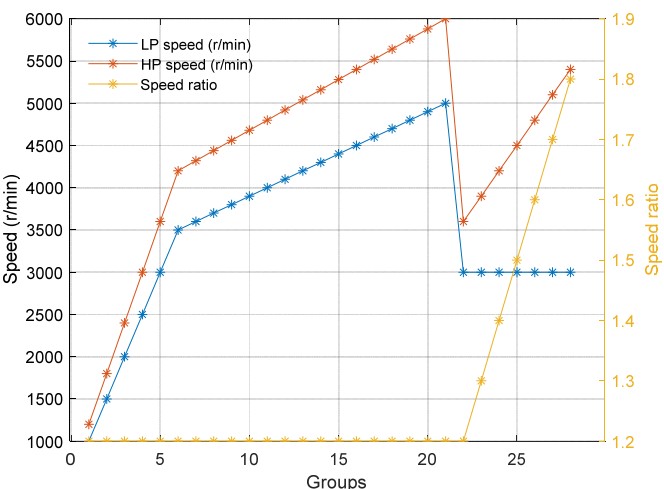

**Figure 2.** Speed test groups for both LP and HP rotors for each of the five subsets.

The vibration signals illustrated in Figure 3 are the outcome of this complicated data generation procedure. Figure 3 illustrates a sample waveform per relative time corresponding to the first channel of vibrations measurements (refer to Figure 1c). It also highlights essential statistical properties such as maximum and minimum values (i.e., *Min*, *Max*), standard deviation ($\delta$), and mean ($\mu$). In this scenario, three distinct operating modes were provided. The information of Figure 3 allows us to basically visualize the degree of difficulties associated with recognizing distinct operating modes. The near-random waveform presentation of the signals is plainly evident. The introductory study gives additional support for this aspect, saying that properties of various operating modes of the inter-shaft bearing cannot be easily detected from the frequency spectrum (see Figure 17 from [8]). This is owing to the fact that that these signals include noise induced by signal distortion during transmission and other frequency components. Due to the apparent difficulties of the phenomena of the harmonic leakage of diagnostic states, frequency analysis is not much useful in this case.

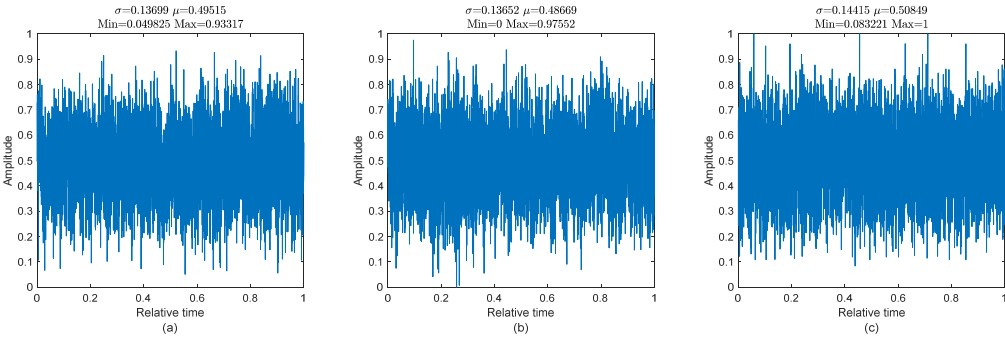

**Figure 3.** A waveform sample of the dataset (i.e., channel 1): (**a**–**c**) waveform signal of healthy, inner ring fault, and outer ring fault operating modes, respectively.

## 2.2. Preprocessing Methodology

As previously indicated in the contributions' subsection, the goal from a well-structured preprocessing methodology is to reduce data complexity while trying to provide meaningful representation to the feature space. This also includes the process making different patterns distinguishable by the learning model via providing better and smoother convergence of the loss function during supervised learning. The mission mainly is to uncover the different operating modes (i.e., healthy and faulty with different types) from the very complex feature space of sensors measurements under massive distortions. Accordingly, our contribution in data preprocessing step is demonstrated by the flowchart diagram of Figure 4. To easily understand different steps of the data preprocessing, Figure 4 introduces them in the form of layers, where some of them could be repeated through the process. The order of these layers is very important as it is defined based on authors expertise in the field repeating different experiment several times with similar classification problems. This subsection is therefore introducing these layers with required details and explanations including a set of very important illustrative examples at the end.

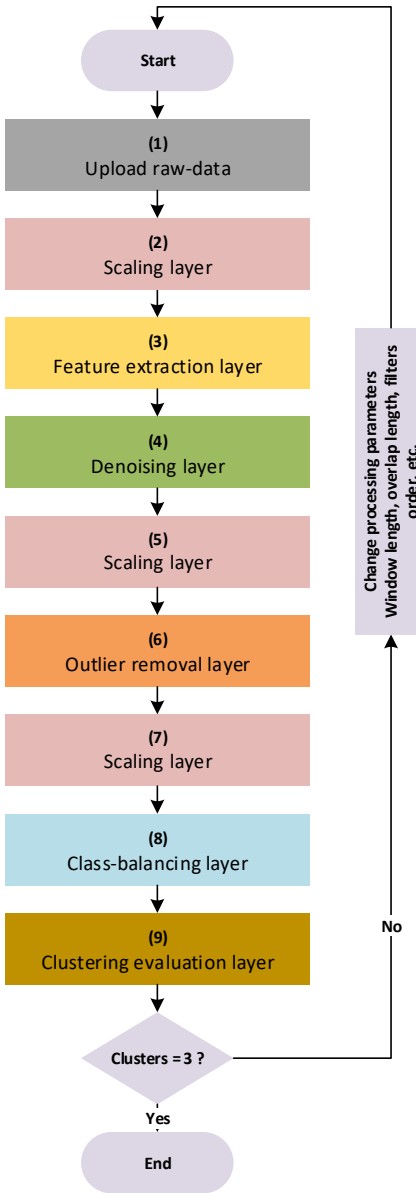

**Figure 4.** Proposed data preprocessing methodology monitored by gaps clustering statistics.

### 2.2.1. Data Uploading Layer

In this paper, data is uploaded chunk-by-chunk in a loop for each preprocessing step using a sliding window according to specific set of parameters of window length, overlapping size, and uniform subsampling rate. These are adjustable parameters controlled by results of the final layer related to gaps analysis. The parameters are considered useful only when optimal number of scatters in prepared data is equal to 3. In this work, these parameters are fixed to 2500, 20, and 0.8, respectively. These parameters are helpful in studying correlation between different time-windows using an overlap while sampling rate reduces data complexity in terms of dimensions and computational time for next layers.

### 2.2.2. Scaling Layer

The scaling layer consists of a set of algorithms dedicated to normalization, smoothing and filtering. These techniques aim to improve the signals quality of all recorded channels as a primary step of data preprocessing paving the way for next layers to further produce better clarity, and usefulness of the signal. This is achieved by eliminating unwanted noise, normalizing its scale, and reducing fluctuations or variations that may hinder analysis or interpretation. The layer is built upon three main algorithms. First, a single-dimensional third-order median moving filtering (MMF) with a fixed time moving window size is used to replaces each sample of collected data in the vibration signals with the median value of its neighboring samples, thus reducing effect of unwanted noise [9]. After filtering, a second slice of smoothing-based moving average filtering (MAF) with fixed window length is also used for purposes of enhancing signal quality and further reduction of noise [10]. Finally, a min-max normalization in the range $[0, 1]$ was used to ensure that all features contribute equally to the learning process and prevent any dominance of certain features based on their original scale [11].

### 2.2.3. Feature Extraction Layer

The performance of the learning model in such classification problem may be influenced by a number of statistical parameters of the input vibration signals. The mean, variance, skewness, kurtosis, autocorrelation, amplitude distribution, and time-domain statistical features are a few of these parameters. Therefore, it is significant to remember that the precise effect of these statistical parameters on the performance of the learning model may differ based on the features of the dataset and the particular classification issue. To ascertain the most important parameters for a given task, a complete list of 15 time-frequency domain features is extracted per each time window (see Section 2.2.1). This layer in fact, provides a way to transform raw signals or data into a more informative and suitable representation for machine learning algorithms. It enables the models to analyze temporal and spectral patterns, handle non-stationarity, and reduce dimensionality, resulting in improved performance and better predictions. Introducing the math behind these features specifically and their significance in terms of diagnostics studies is not the main goal of this paper as they are well-familiar in state-of-the-art literature. However, further mathematical details and an in-depth analysis of these features can be found in [12], while these features are listed from 1 to 15 in Appendix A of the provided reference.

### 2.2.4. Denoising Layer

This work adopts the use of Wavelets with Cauchy priori and the Bayesian method in denoising the extracted features from the non-stationary vibration signals of the second preprocessing layer [13]. These techniques offer an effective way to remove noise from signals while preserving the important features of the original signal. The combination of wavelets, Cauchy priori, and the Bayesian method provides an effective approach for denoising non-stationary vibration signals. These techniques allow for a better representation of the signal in both time and frequency domains, resulting in enhanced denoising performance compared to traditional linear filtering methods and those presented early in the scaling layer. It should be mentioned that there is possibility of the denoising layer

to change data features scale by reducing amplitudes of harsh noise. Therefore, it will be beneficial for the scaling layer to be reused in this stage to rescale data again.

### 2.2.5. Outlier Removal Layer

Outliers arising due to various reasons, such as sensor malfunctions, measurement errors, or rare events can distort the statistical properties of vibration signals. Machine learning algorithms heavily rely on these statistical properties for reliable learning and predictions. By removing outliers, data can better represent the underlying distribution, leading to more accurate fault diagnostics. Outliers can have a disproportionate impact on the model training process. Machine learning algorithms, especially those based on distance metrics or clustering, can be heavily influenced by the presence of outliers. Removing these outliers helps in minimizing their influence and improves the overall robustness and performance of the model. In this work, a list of outlier detection methods is involved in a single layer. These methods can be classified into statistical, distance-based, density-based, and cluster-based methods [14] including 8 different methods namely; median analysis, mean analysis; quartiles analysis, Grubbs test, generalized extreme studentized deviate, Mahalanobis distance, Euclidean distance, Minkowski distance. The variety of methods is involved to make sure that outliers are reduced as much as possible from provided signals. After that, another scaling layer is necessary to be added with the aim of adjusting features scale again before next steps.

### 2.2.6. Class-Balancing Layer

Synthetic minority oversampling technique (SMOTE) is used in this work to address the problem of imbalanced datasets. In fault diagnosis, there may be instances where the number of samples in the minority class (faulty instances) is significantly lower than the number of samples in the majority class (normal instances). This imbalance may result in biased models that have poor performance in detecting faults. Further details background of this method can be found in [15]. In the current preprocessing scheme and uploading layer settings, a 130,011 $\times$ 15 matrix representing observations from the dataset where the class proportions are equally distributed is obtained as a final dataset for training and validation.

### 2.2.7. Clustering Evaluation Layer

This layer uses a Gap clustering metric to evaluate data quality resulted from all preprocessing steps. Gap clustering evaluation can be used for data quality assessment by measuring the quality of a dataset based on its cluster ability. This technique evaluates the effectiveness of clustering algorithms in separating the data points into distinct clusters. Gap clustering evaluation can be used for data quality assessment by selecting a clustering algorithm, generating multiple clustering solutions, computing and comparing Gap statistics, determining optimal number of clusters, and assessing data quality for supervised learning. This process allows identifying patterns, validating assumptions, and gaining insights about the dataset structure [16]. In this work, k-means method is involved as primary step of evaluation in unsupervised learning process. Optimal number of clusters is computed and compared to raw dataset scatters analysis results accordingly. Consequently, if optimal number of clusters is the same as data classes (i.e., 3 clusters in the studied dataset), the data preprocessing considered acceptable, otherwise the process should be repeated with other data uploading parameters until reaching satisfactory results. I such case, one might think that clustering algorithms such as k-means, with their ability to uncover inherent structures in data, could suffice for fault diagnosis. However, the decision to opt for LSTM in our study is driven by specific requirements of fault diagnosis tasks. While clustering indeed provides valuable insights into the data structure and aids in initial data quality assessment, it may not inherently improve data representations in a manner conducive to the intricacies of fault diagnosis. LSTMs, on the other hand, are specifically designed to capture complex temporal dependencies and sequential patterns within data.

Their end-to-end learning capability, nonlinear pattern recognition, and effectiveness in handling sequential information make them a more suitable choice for fault diagnosis tasks. Therefore, the aim is to leverage both clustering, for initial data quality assessment, and deep learning, particularly LSTM, for a more comprehensive approach to fault diagnosis. This ensures not only the identification of clusters but also the learning and generalization from intricate patterns present in fault scenarios.

### 2.3. Some Illustrative Examples

Table 2 presents a comprehensive summary of the calibrated parameters used in the process for data quality enhancement.

**Table 2.** Parameters of data quality enhancement layers.

| Layers | Options |
|---|---|
| Upload raw-data | <ul><li>Time window length = 2500 samples;</li><li>Overlap = 20 samples.</li></ul> |
| Scaling layer | <ul><li>MMF span = 5 samples;</li><li>MAF span = 3 samples;</li><li>Min-max normalization interval = [0,1].</li></ul> |
| Feature extraction layer | <ul><li>Number of features = 15 (i.e., 11 time-domain features and 4 frequency domain features;</li><li>Time window length = 2500 samples;</li><li>Overlap = 20 samples;</li></ul> |
| Denoising layer | <ul><li>Wavelets type = orthogonal symlets;</li><li>Denoising method = Empirical Bayes;</li><li>Threshold rule = median threshold;</li><li>Method of estimating variance of noise = highest-resolution wavelet coefficient;</li><li>Level of decompositions = $\log_2 N$; $N$ is number of samples.</li></ul> |
| Outlier detection layer | <ul><li>Detectors: median, mean, Grubbs, and quartiles; Mahalanobis, Euclidean, Minkowski distances;</li><li>Node samples = 50 samples;</li></ul> |
| Class-balancing layer | <ul><li>Method = SMOTE;</li><li>Neighbors = 3.</li></ul> |
| Clustering evaluation layer | <ul><li>Clusters = k-means</li><li>Optimal number of clusters = 3;</li></ul> |

Using preprocessing parameters indicated in Table 2, Figure 5 is obtained. Figure 5a,b represent data scatters of raw-data and prepared data, respectively. These scatters are obtained from a 3-dimensional transformation of the 15 extracted features using t-distribution stochastic neighbor embedding (t-SNE) that is commonly used for visualizing high-dimensional data in a lower-dimensional space [17]. In the meantime, Figure 5c represents Gaps statistics obtained using k-means technique for number of scatters equal to 3. By comparing Figure 5a,b, data scatters can be distinguished in the prepared version better than scatters in raw-data. This is supported by agglomeration seen in samples of different classes. Accordingly, these samples are therefore aggregated and located next to each other, which improves patterns separability thanks to the proposed data preprocessing methodology. When it comes to classification process, such enhanced separability is extremely beneficial as it boosts learning models performances. Contrariwise, raw-data scatters are showcased in a sort of single spot difficult to be distinguished from each other. So, contributions of the preprocessing can be clearly seen in such case. This is the reason for using a simple deep learning architect of 10 neurons single LSTM layer model in next sections. Furthermore, for maximum number clusters equal to 3, Figure 5c shows the scattering ability of the dataset to three distinguished groups (i.e., highest Gap value is given to optimal number of clusters). This means the presence of 3 distinguishable types of patterns in the dataset. In the meanwhile, the raw-data elucidates that optimal number of data scatters is 1 due to

complexity and nature of samples merged with outliers and noise. This further clarifies importance and accuracy of data preprocessing scheme.

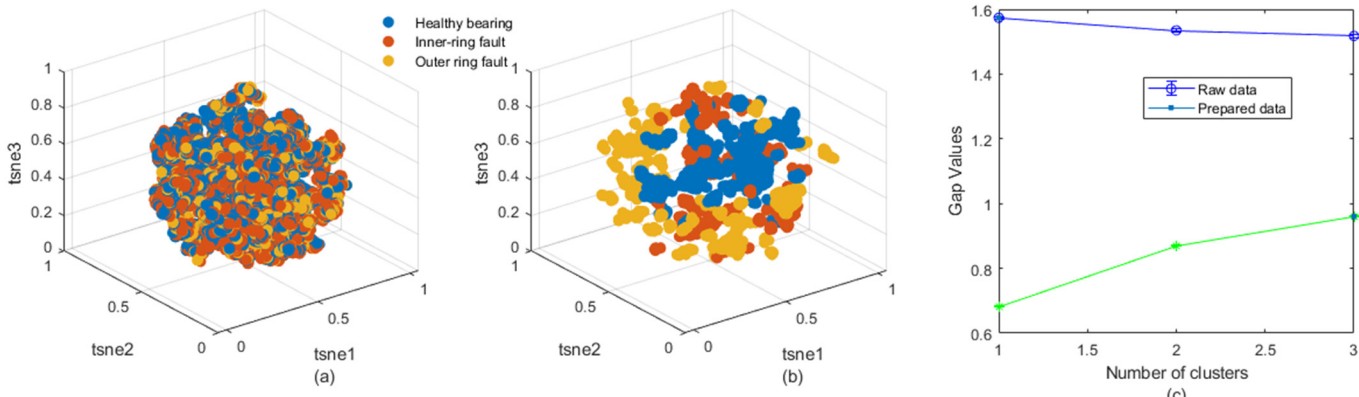

**Figure 5.** Visualization of data scatters versus Gap statistics: (**a**) t-SNE scatters of raw-data; (**b**) t-SNE scatters of prepared data; (**c**) Gap statistics results for both prepared and raw-data.

### 3. Methods and Findings

Preprocessed data according to proposed methodology in Section 2 is fed to the proposed RWI-LSTM approach for further deep learning modeling and evaluation process. In order to make the proposed approach as well as obtained conclusion clearer, this section is subdivided into two subsections where proposed method, application results and discussion are emphasized clearly and separately.

*3.1. Methods*

The proposed RWI method for LSTM improvement is dictated by the diagram of Figure 6. The math behind LSTM and its contribution in adaptive learning is well-known in the literature, as for instance, reference [18] provides such very important details. Our methodology facilitates the recursive initialization of learning weights in various rounds, denoted as $k$. In each round, the LSTM network undergoes initialization with new weights. The selection of these weights is conducted meticulously in accordance with the RWI philosophy. In another way, the LSTM layer is initialized at first round $k = 1$ with input weights $w_k^i$, recurrent weights $w_k^r$ and biases $b_k$ from a specific probability distribution $P$ with mean $\mu$ and standard deviation $\delta$ as in (1). The training process involves cross validation process resulting in a variety of models with different completely tuned parameters $\left\{ w_{(f,\,k)}^i, w_{(f,\,k)}^r, b_{(f,\,k)} \right\}$ with $f$ is the fold index and $m$ maximum number of folds. These parameters will be subject to next selection process based best validation performances and training as in (2). This is done in this case to find the best models for delivering improved generalization on new unseen samples. The term generalization in this work refers to the capacity of the learning model to perform well on previously untrained, novel, or unseen data. The RWI process will be repeated for any rounds $k$, while the best results could appear at any of the rounds. As a result, the more rounds we have, there is possibility to stack in better learners. The next subsection will explain this methodology and provide further insights about the training process while also discussing limits of the methodology.

$$\left\{ w_k^i, w_k^r,\, b_k \right\} = P(\mu, \delta) \tag{1}$$

$$\left\{ w_{k+1}^i, w_{k+1}^r,\, b_{k+1} \right\} = \left\{ w_{(f_{best},\,k)}^i, w_{(f_{best},\,k)}^r, b_{(f_{best},\,k)} \right\} \Big| \\ Accuracy_{f_{best}} = \max \left( Accuracy_{(f,k)} \right)_{f=1}^{m} \tag{2}$$

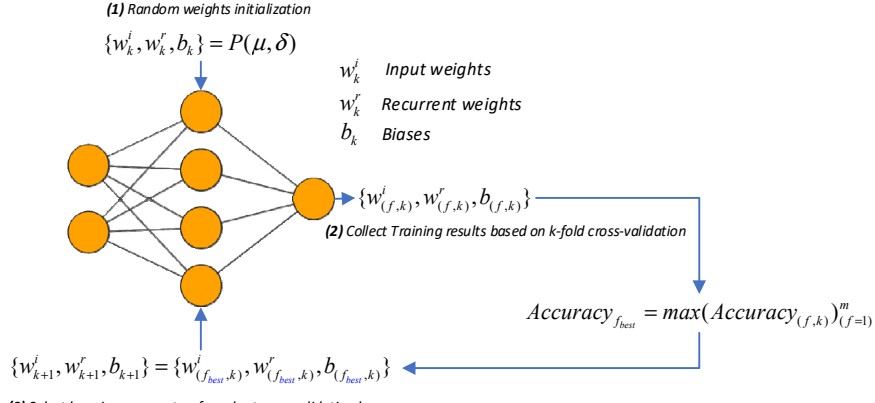

**Figure 6.** Schematic diagram of the proposed RWI approach.

It is reasonable to think of RWI as an optimization process similar to starting with a random weights initialization and employing gradient descent to converge to a potential local solution. This analogy holds true when we disregard the selection process of the best learners from the k-folds cross-validation models. However, introducing this selection process across multiple rounds is crucial, as it ensures the reproduction of only the most useful solutions.

### 3.2. Application, Results and Discussion

The RWI in this work is fixed to 3 folds for cross validation parameters, and 40 rounds of training. This means that about $40 \times 3$ learning models and confusion matrices are investigated. The LSTM network in this case uses only a single LSTM layer with 10 neurons while the other parameter tuning process is inspired from original work in [8]. The training and evaluation process took 1.1177 h while main classification metrics include *Accuracy*, $F1 - score$, *Precision* and *Recall* widely used in the literature (for further details about these metrics and their significance please refer to [19]).

First, Figure 7 shows the achieved results centered on the testing phase, while the provided metrics results are the average value of the results obtained in each of the three cross validation folds at each round. In this curve, and by observing the peaks in performances, it can be seen that RWI helps in improving classification performances gradually by each round showing a reasonable increase. This uncovers one of main advantages of this methodology in parameter initialization. Contrariwise, when RWI passes round 13, model performances start deteriorating and such a process becomes no longer effective. This suggests that the RWI process should be controlled as it could bias the model when overpassing the maximum required number of rounds.

Second, illustration of the loss function and classification accuracy behavior during the training process of the best learners' performances in round 1 (i.e., ordinary LSTM), and of RWI-LSTM at round 13 could give further details about the performances of the proposed methodology. One might think that LSTM with its current structure could have a negative impact since it is used after reduced dimension feature extraction. However, even after such a process, LSTM is still necessary because it can capture long-term dependencies and patterns in data. All that is required is the extracted features' necessity to have a sequential perspective and maintain their order. This work uses LSTM to learn and predict sequential patterns in serialized data by enabling time window extraction, taking into account overlaps while preserving sequential behavior. According to the authors expertise in similar topics [20], the LSTM structure does not necessarily have a negative impact on the classification results because it is often used in sequence classification tasks. Other factors that contribute to lower classification performance may include inadequate hyperparameter tuning and data complexity. Nevertheless, an additional comparison with a traditional single-layer feedforward network (SLFN) with

the same parameters will be of great advantage in confirming this claim. Hence, Figure 8 is inserted for this purpose. The behavior of the loss function in Figure 8a during the training phase contributes to demonstrate another LSTM advantage brought forth by the RWI technique, namely a faster convergence process. Even with the presence of few fluctuations at the beginning, faster and more accurate stability is achieved quickly compared to ordinary LSTM. It seems that further stability can be gained by increasing the number of iterations in this case. However, similar learning parameters of the work introduced in [8] are kept in an attempt to achieve faire comparisons. In contrast to RWI-LSTM and LSTM, the loss function behavior of SLFN exhibits a poor starting and convergence regime. This demonstrates unequivocally why we decided to use an adaptive learner that is updated constantly in response to changes in the data. Likewise, similar findings also apply to the analysis of Figure 8b.

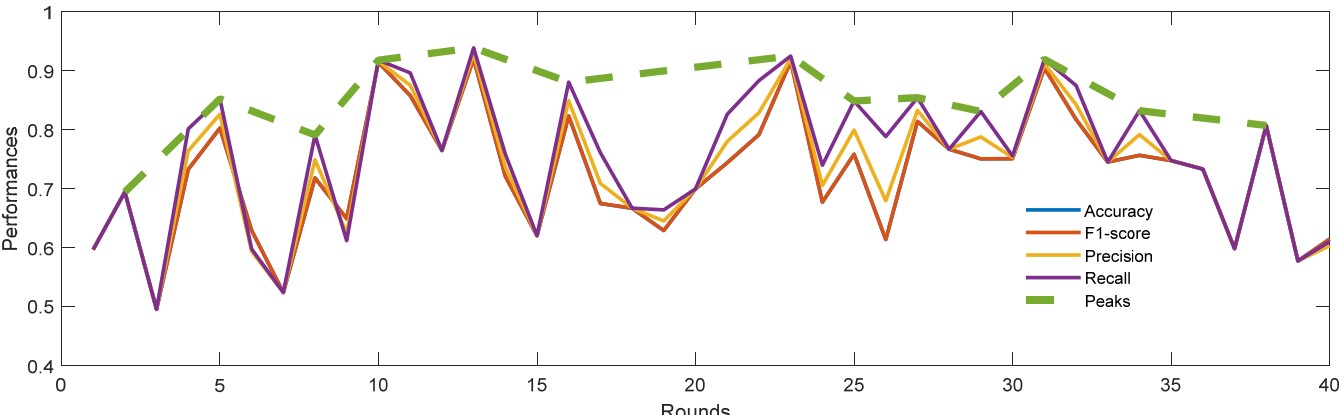

**Figure 7.** A summary of results obtained during entire learning process.

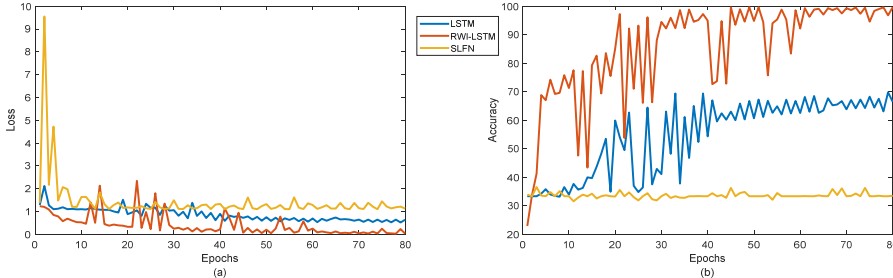

**Figure 8.** Loss and accuracy comparison: (**a**) loss function behavior; (**b**) classification accuracy behavior.

As previously illustrated, the best obtained results in this case are those of round 13. Accordingly, if we compare results obtained from this work to LSTM networks, and work's results originally published in [8], Table 3 results are the finally obtained numerical outcomes dedicated to comparisons and assessment of learning model performances. The results clearly highlight robustness of the proposed methodology compared to existing approaches in the literature with a very small standard deviation. Clearly, tackling data preprocessing with proposed methodology and involving a better RWI, allows the LSTM network to reach satisfactory results even with a small number of neurons (i.e., 10 neurons) less than the number of neurons in the input layer (i.e., 15 neurons).

**Table 3.** A recapitulation of the obtained results in terms of various performance criteria.

| Method | *Accuracy* | *F1 − Score* | *Precision* | *Recall* | Standard Deviation | Evaluation Method |
|---|---|---|---|---|---|---|
| SLFN * | 0.3438 | 0.3438 | 0.3438 | 0.3438 | $10^{-8}$ | Cross validation |
| LSTM * | 0.596 | 0.590 | 0.596 | 0.596 | $1.77 \times 10^{-5}$ | Cross validation |
| LSTM [8] | 0.854 | - | - | - | - | Random 70–30% splitting |
| RWI-LSTM * | **0.920** | **0.920** | **0.929** | **0.938** | $\mathbf{8.4901 \times 10^{-4}}$ | Cross validation |

* Experiments performed in this paper.

*3.3. Comparison Statement*

It is crucial to note that results obtained in this study are derived through the implementation of cross validation techniques, ensuring a more robust and reliable evaluation of the RWI-LSTM performance. This approach provides an unbiased estimate of generalizability to unknown data. In contrast to previous studies employing the random 70–30% technique, our model demonstrates superior generalization, a facet that was uncertain in earlier works. Furthermore, our study addresses the complexities of data and drift problems through a well-structured approach. The inclusion of data preprocessing as a primary step allows for the discernment of different data scatters, offering a distinct advantage over existing works that overlook this aspect. In a more specific context, the work referenced in [8] employs a realistic dataset and utilizes the LSTM method. However, our study surpasses this by incorporating an improved LSTM with RWI, going beyond the limitations of a standard LSTM. Notably, our research addresses the challenges associated with data quality enhancement, a dimension not explored in the original reference [8]. These assertions are explicitly detailed in Table 1 of related works analysis and in Section 1.1 before elucidating the research gaps. Additionally, it is noteworthy that the evaluation strategy in reference [8] relies on a random splitting approach, whereas our work adopts a more robust cross validation methodology. Consequently, it should be emphasized that comparisons presented in Table 3 are not only unfair but also lack robustness. The enhanced robustness and trustworthiness of our proposed method are further underscored when considering these critical criteria.

**4. Conclusions**

This work introduces a comprehensive methodology aimed at gaining a deeper understanding of both data complexity and drift in highly dynamic safety-critical aircraft engines. The methodology addresses the challenge of fault diagnosis under the utilization of highly non-stationary vibration signals, with a specific focus on inter-shaft bearings as a case study. Two primary phases, involving proper data engineering and adaptive deep learning with improved RWI, are employed to address the complexities and drift inherent in the data. In the first phase, data quality is assessed using gap statistics, while the second phase, involving the learning process, is evaluated using well-established metrics. Cross validation plays a crucial role in this assessment, serving as a primary tool for learning process evaluation. The application results, when compared to previous works, reveal significant insights into the necessity of combining data quality and an accurate learning process. Looking towards future opportunities, advancements in this field will persist by addressing two key aspects. First, in the realm of data engineering, further exploration of denoising, outlier removal algorithms, and general data complexity reduction techniques is warranted, accompanied by the introduction of additional methods for assessing data quality. Second, the deep learning architecture should be enhanced by incorporating additional automatic extraction and abstraction layers to reveal more meaningful representations. Furthermore, from a decision-making perspective, a third aspect deserves consideration. Future opportunities can delve into the preemptive capabilities of such techniques and their potential impact on supporting pilot decision making to mitigate potential risks. We

believe that by concentrating on the early detection of faults, our technique can significantly contribute to strengthening the safety criteria of flight operations.

**Author Contributions:** Conceptualization, T.B. (Tarek Berghout); methodology, T.B. (Tarek Berghout) and M.B.; validation, T.B. (Tarek Berghout) and M.B.; formal analysis, T.B. (Tarek Berghout) and M.B.; investigation, T.B. (Tarek Berghout); resources, T.B. (Tarek Berghout); data curation, T.B. (Tarek Berghout); writing—original draft preparation, T.B. (Tarek Berghout); writing—review and editing, T.B. (Tarek Berghout), T.B. (Toufik Bentrcia), W.H.L. and M.B. All authors have read and agreed to the published version of the manuscript.

**Funding:** This research received no external funding.

**Data Availability Statement:** The source code, along with all the essential materials and information necessary to replicate the findings of this study, can be accessed at: https://doi.org/10.5281/zenodo.10184658.

**Acknowledgments:** The authors would like to thank the Harbin Institute of Technology, and the authors of the introductory article Hou, L.; Yi, H.; Jin, Y.; Gui, M.; Sui, L.; Zhang, J.; Chen, Y.; for making their dataset available, which proved very important and useful for conducting the proposed methodology.

**Conflicts of Interest:** The authors declare no conflict of interest.

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
