# Peer review of "A Neural Network Weights Initialization Approach for Diagnosing Real Aircraft Engine Inter-Shaft Bearing Faults"

_machines, doi:10.3390/machines11121089_

Round 1

Reviewer 1 Report

Comments and Suggestions for Authors

This work introduced a method of fault diagnosis in safety critical aircraft engines, which includes two parts: feature extraction and adaptive diagnosis, and has achieved good classification results. Compared with other studies, this work used a relatively new data set and data processing method. However, there are still defects in the content arrangement of the manuscript.

1.     There may be spelling errors in the text, such as line 14 of the second paragraph of section 1.1, page 2. Please carefully check the relevant work.

2.     The description of the operating environment and file format in Section 2.2 is too redundant, and it is recommended to delete it.

3.     There are confusing parts in the description of Figure 4 in Section 2.3. First of all, the original data set described in Section 2.1 includes three failure modes and two health modes, a total of five modes, but Section 2.3 only carries out the test of three categories. Through consulting the corresponding references, it is understood that the original data and indeed only cover three fault states, but the description of the corresponding part of the text should also be modified accordingly to reduce possible obstacles to understanding. And, it is suggested to mark the fault type corresponding to the data in the legend of (a)(b) subfigure in Figure 4, instead of directly showing the data label. In addition, the result of feature extraction shown in Figure 4 (b) is not ideal, and Class 0 features are almost invisible. Can the author consider using other data dimensionality reduction methods to better demonstrate the effect of feature extraction?

4.     In this paper, LSTM is used to classify the data set after feature extraction, but is it still necessary to use LSTM for the data after dimension reduction? Does the extracted feature meet the needs of LSTM for serialized data? Starting from the test results of using LSTM alone, it is reasonable that the structure of LSTM itself has a negative effect on the classification results. Can the author give the classify accuracy of traditional neural networks with similar parameter scale for reference?

5.     The generalization ability of the model is emphasized in Section 3.1. Is this generalization ability for which case, working conditions or system dynamics characteristics? Please provide additional instructions in the text.

Comments on the Quality of English Language

The English should be polished.

Author Response

Dear Associate Editor and Reviewers,

The authors are thankful for the efforts of the Editor and the Reviewers for the evaluation of our proposal. The relevant comments and suggestions have definitely improved both the proposal presentation and quality. In this context, the authors have incorporated all the suggestions in the revised manuscript and have addressed all the raised issues. In this document, we propose detailed responses to the comments and questions. All the revisions are highlighted in red in the revised manuscript.

With Regards

Reviewer 2 Report

Comments and Suggestions for Authors

The paper studies data methods for diagnosis of aircraft engine, which is maybe useful for fault diagnosis and detection of engines. However, the whole paper makes readers confusing:

-. What kinds of engine data are used? And where are they from?

-. What are the real problems this paper wants to solve?

This version makes me thinking just use some engine data to do data analysis, and it is cannot be published.

Author Response

Dear Associate Editor and Reviewers,

The authors are thankful for the efforts of the Editor and the Reviewers for the evaluation of our proposal. The relevant comments and suggestions have definitely improved both the proposal presentation and quality. In this context, the authors have incorporated all the suggestions in the revised manuscript and have addressed all the raised issues. In this document, we propose detailed responses to the comments and questions. All the revisions are highlighted in red in the revised manuscript.

With regards,

Reviewer 3 Report

Comments and Suggestions for Authors

Comments and Suggestions for Authors:

The authors presented in the paper the Network Weights Initialization Approach in order to train the LSTM type of deep neural networks under the problem of the real time diagnosis systems. Currently, the subject matter presented in the work is discussed by many authors, that is why in my opinion the paper is interesting and the topic fits in general the scope of the journal well.

Although the main idea of the article is clear, in my opinion, the article itself does not explain precisely how the proposed recursive weighting methodology could be used in other real systems that enable intelligent engine diagnostics. Input preprocessing is a key module for many real-world diagnostic systems, so the correct configuration and operation of which have a critical impact on the efficiency of the entire system. Considering the above, I invite the authors to discuss and formulate additional clarifications.

1.       I recommend to the authors of the paper to include in the section devoted to the analyzed signals, sample waveforms of vibrations, a summary of statistical properties for the correct and faulty operation of the adopted power unit. This will make it possible to estimate what is the degree of difficulty of the task being solved and whether the phenomenon of harmonic leakage between diagnostic states appears in the process. For clarity, the time step can be selected so that those fragments that are crucial for the parameters of the whole process are visible.

2.       Figure 2 does not explain clearly enough what the speed test groups introduced mean. Why was this number of them adopted?

3.       Figure 3 shows the adopted feature engineering methodology. What are the values and ranges for manipulating the Window length, overlap length, filters parameters? As it stands, the reader can only get an approximate idea of the actual scale of the equilibrium sought. The lack of the precise constraints means that in the theoretical case the complexity of the algorithm can be unlimited.

4.       Many studies on network learning strategies present initial and final weight sets that indicate the intended and achieved performance of the network. Therefore, I suggest to authors to visualize the best and worst weight sets of LSTM networks that were achieved using the RWI method. Which of the statistical parameters of the input signal, in authors opinion, can affect the results of LSTM from using RWI?

5.       Sugeruję aby autorzy posługiwali się pojęciem data preprocessing (tak jak w linii 349) zamiast data processing (linie 333, 335 itd.), które wskazuje na metody i początkowy etap przetwarzania danych przed procesem uczenia.

6.       Figure 4 b) in my opinion does not explain the differences in visualization with sufficient precision. Please describe in detail in the text the interpretation of part c of this figure. I suggest supplementing Figure 7 with a graph of Accuracy(Iteration).

7.       I suggest to change the global title of the chapter Methods, application, results and discussion.

8.       In the final part of the paper, I encourage the authors to discuss the effectiveness of the proposed method in the task of detecting anomalous operation of aircraft engines. Please note that the detection of defects, for example, during flight, does not change the safety parameters of flight. It is much more desirable to detect their first symptoms or anomalies, which preemptively support the pilot's work and enable avoidance of potential hazards.

Formatting, specific comments:

Line 28 only10

Line 359 – distribution𝑃,

Author Response

(The authors gave the same response as above.)

Round 2

Reviewer 2 Report

Comments and Suggestions for Authors

The authors have made a good revision by the first round comments. A minor issue is that the conclusion section needs to be written more specific and carefully. The present one is too subjective and the description is too strong, such as "an entire methodology", "all challenges", etc.

As data methods were used for diagnosis of aircraft engine, and were useful for fault diagnosis and detection of engines, the conclusion section of this paper should give keys from present results.

Author Response

Dear Associate Editor and Reviewers,

The authors are thankful for the efforts of the Editor and the Reviewers for the evaluation of our proposal. The relevant comments and suggestions have definitely improved both the proposal presentation and quality. In this context, the authors have incorporated all the suggestions in the revised manuscript and have addressed all the raised issues. In this document, we propose detailed responses to the comments and questions. All the revisions are highlighted in red in the revised manuscript.

Kind regards
